# HERD: Continuous Human-to-Robot Evolution for Learning from Human Demonstration

**Xingyu Liu**     **Deepak Pathak**     **Kris M. Kitani**
Carnegie Mellon University

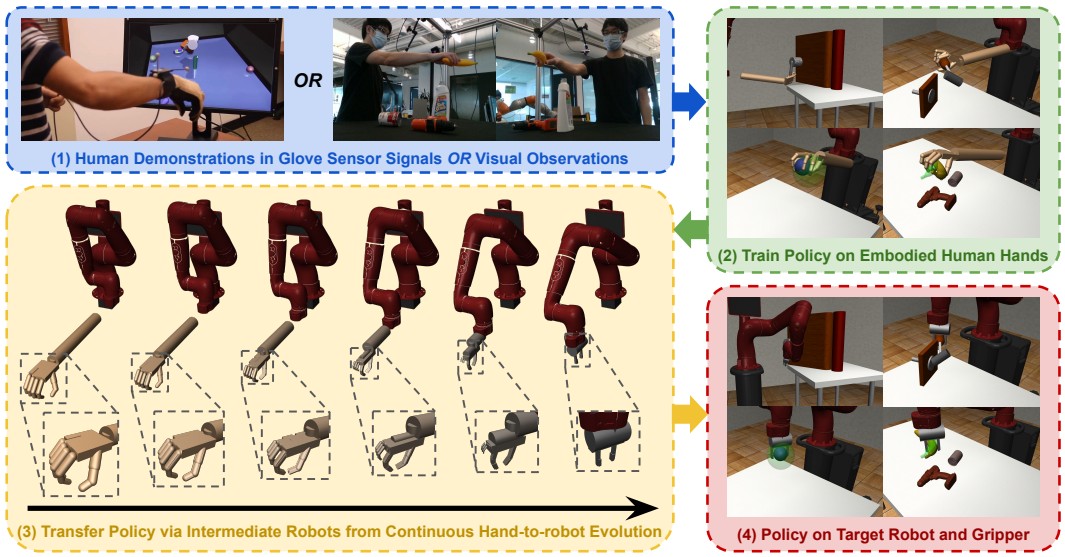

Figure 1: **HERD framework:** (1) Given human demonstrations in sensor glove signals [1] or visual data [2], (2) expert policies on an embodied human hand, i.e. a five-finger dexterous robot, can be trained. (3) We then design continuous robot evolution to connect the five-finger dexterous robot and a target commercial two-finger-gripper robot. A smooth curriculum of policy optimization on the intermediate robots that gradually evolve towards the target robot allows (4) the expert policies to be transferred to the target robot even in challenging sparse-reward tasks.

**Abstract:** The ability to learn from human demonstration endows robots with the ability to automate various tasks. However, directly learning from human demonstration is challenging since the structure of the human hand can be very different from the desired robot gripper. In this work, we show that manipulation skills can be transferred from a human to a robot through the use of micro-evolutionary reinforcement learning, where a five-finger human dexterous hand robot gradually evolves into a commercial robot, while repeated interacting in a physics simulator to continuously update the policy that is first learned from human demonstration. To deal with the high dimensions of robot parameters, we propose an algorithm for multi-dimensional evolution path searching that allows joint optimization of both the robot evolution path and the policy. Through experiments on human object manipulation datasets, we show that our framework can efficiently transfer the expert human agent policy trained from human demonstrations in diverse modalities to target commercial robots.

## 1   Introduction

Learning from human demonstrations [3] is a promising direction to enable robots to perform diverse manipulation capabilities. Recent large-scale visual datasets of human activity [2, 4, 5, 6, 7] have highlighted the need for enabling robotic agents to learn from human activities to perform diverse tasks in real-world environments. Existing paradigms for learning from human demonstration

6th Conference on Robot Learning (CoRL 2022), Auckland, New Zealand.

generally follow the pipeline of ***demonstration conversion***, i.e. (1) ***convert demonstrated human states to robot states*** and then (2) ***train the robot policy***. Though step (2) can be solved by existing imitation learning approaches [8, 9, 10, 1], step (1) is extremely difficult. Efforts to address step (1) include human-robot observation matching such as [11], and converting human demonstrations to robots by human-robot interaction [12, 13] or human-in-the-loop teleoperation [14, 15]. However, these solutions are highly situational and task-specific and require significant human intervention for each individual task, therefore are not scalable and cannot automatically generalize to new tasks.

We propose a new paradigm for learning from human demonstration — ***policy transformation***, i.e. (1) ***train policy on embodied human hand dexterous robot*** and then (2) ***transfer the policy from dexterous hand robot to the target robot***. Many existing methods have been shown successful in learning control policy for dexterous hand robots from human demonstration [1, 16] and can easily solve step (1) of our pipeline. A general and automatic solution of step (2) would render a scalable solution of learning from human demonstration. The key challenge to step (2) is the huge dynamics discrepancy between the dexterous hand robot and the target robot. The discrepancy stems from the mismatch in robot morphology and kinematics. Unfortunately, statistical matching imitation learning approaches such as [8, 9, 10, 17] assume the teacher and student robotic agents share the same or similar transition dynamics, therefore are ineffective in solving the problem.

In this paper, we present a framework named *HERD* (short for *H*uman *E*volution to *R*obot for Learning from *D*emonstration) with a new perspective on solving the problem of learning from human demonstration — ***continuous robot evolution***. Following the philosophy introduced in the seminal work of REvolveR [18], the core idea of our framework is to define a continuous interpolation between the dexterous hand robot and a target robot that have different morphology. Then an expert dexterous hand robot policy learned with methods such as [1, 16] can be transferred to the target robot through training on a sequence of intermediate robots that gradually evolve into the target robot. The policy is continuously updated through repeated interaction in a physics engine that contains the intermediate robots, as illustrated in Figure 1.

To instantiate the idea, we develop two solutions for the evolution from five-finger dexterous robot to two target commercial robots respectively: a Sawyer robot with a two-finger Rethink gripper, and a Jaco robot with a three-finger Jaco gripper. The solutions precisely match the dynamics of target commercial grippers. To deal with the high dimensions of robot parameters, we propose an algorithm for automatically searching evolution paths in high-dimensional robot parameter space that allows joint optimization of both robot evolution and the policy, based on online estimation of reward gradient with respect to robot evolution.

We conduct experiments on transferring human expert demonstrations of manipulation tasks in diverse modalities to the target commercial robots. We show that HERD can successfully transfer the expert dexterous hand robot policy trained from Hand Manipulation Suite sensor glove demonstrations [1] to the target robots even with sparse rewards. On DexYCB dataset [2], we demonstrate that our approach can also transfer the expert policy trained with visual human demonstrations to the target robots. In addition, we show that the proposed multi-dimensional robot evolution path search algorithm can significantly improve the policy transferring efficiency.

## 2 Preliminaries

**MDP Preliminary**  We consider a continuous control problem formulated as Markov Decision Process (MDP). It is defined by a tuple $(\mathcal{S}, \mathcal{A}, \mathcal{T}, R, \gamma)$, where $\mathcal{S} \subseteq \mathbb{R}^S$ is the state space, $\mathcal{A} \subseteq \mathbb{R}^A$ is the action space, $\mathcal{T} : \mathcal{S} \times \mathcal{A} \to \mathcal{S}$ is the transition function, $R : \mathcal{S} \times \mathcal{A} \to \mathbb{R}$ is the reward function, and $\gamma \in [0, 1]$ is the discount factor. A policy $\pi : \mathcal{S} \to \mathcal{A}$ maps a state to an action where $\pi(a|s)$ is the probability of choosing action $a$ at state $s$. Suppose $\mathcal{M}$ is the set of all MDPs and $\rho^{\pi,M} = \sum_{t=0}^{\infty} \gamma^t R(s_t, a_t)$ is the episode discounted reward with policy $\pi$ on MDP $M \in \mathcal{M}$. The optimal policy $\pi_M^*$ on MDP $M$ is the one that maximizes the expected value of $\rho^{\pi,M}$.

**REvolveR [18] Preliminary**  Liu et al. [18] proposed a technique named REvolveR for transferring policies from a source robot to a target robot. The core idea is to define an evolutionary sequence of intermediate robots that connects the source to the target robot. Given source and target robots represented by two MDPs $M_S, M_T \in \mathcal{M}$ respectively, REvolveR defines a continuous function $E : [0, 1] \to \mathcal{M}$ where $E(0) = M_S, E(1) = M_T$. Then an expert policy $\pi_{E(0)}^*$ on the source robot $E(0)$ is optimized by sequentially interacting with each intermediate robot in the sequence

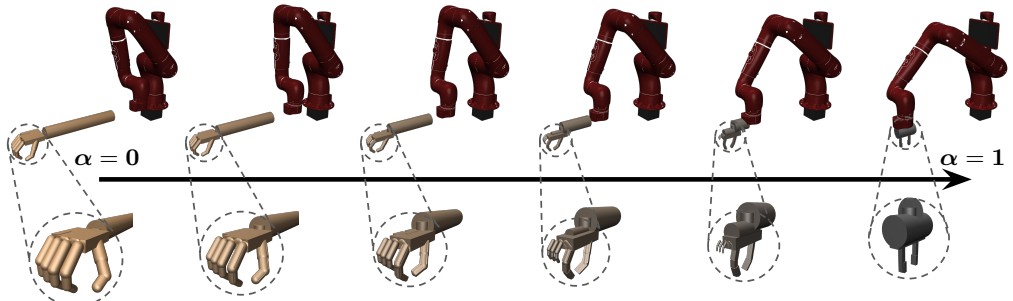

Figure 2: **Continuous hand-to-robot evolution for Sawyer robot.** We show one evolution path from human dexterous hand ($\alpha = 0$) to a commercial Sawyer robot with two-finger Rethink gripper ($\alpha = 1$). From left to right, we show intermediate robots along the path. Zoom in for better view.

$E(\alpha_1), E(\alpha_2), \ldots, E(\alpha_K)$ where $0 < \alpha_1 < \alpha_2 < \cdots < \alpha_K = 1$, until the policy is able to act (near) optimally on each intermediate robot and eventually transfer the policy to target robot $E(1)$. For all $i$, $|\alpha_{i+1} - \alpha_i|$ is small enough so that each policy fine-tuning step is a much easier task. Our method is built upon the philosophy of REvolveR to use robot evolution for policy transfer.

**Notation** We use bold letters to denote the vector variables. Specially, $\mathbf{0}$ and $\mathbf{1}$ are the all-zero and all-one vectors with proper dimensions respectively. $S^{D-1}(\xi) = \{\mathbf{x} \in \mathbb{R}^D \mid ||\mathbf{x}||_2 = \xi\}$ is a $(D-1)$-sphere (i.e. a sphere in $\mathbb{R}^D$ space) with radius $\xi \in \mathbb{R}^+$. $\odot$ is element-wise product.

## 3 Method

### 3.1 Human Demonstration to Dexterous Robot Expert Policy

Recent works [1, 16] have shown success in learning five-finger dexterous robot policy from human hand demonstrations in diverse modalities, such as sensor glove signals [1] and visual data [16], which made the first step of our proposed paradigm possible. Inspired by this research progress, we adopt similar approaches to train expert policies on a dexterous robot as illustrated in Figure 1(1)(2).

**Sensor Glove Demonstration** When recorded in sensor glove signals, human demonstrations are directly converted to states and actions in a simulator [19] from which a policy is trained with algorithms such as DAPG [1]. We use the same procedure and obtain the expert policy as in [1].

**Visual Demonstration** We develop a toolchain for training dexterous policies from human visual demonstrations of manipulation. Our toolchain includes human and object pose estimation with visual perception followed by pose fitting of the dexterous robot via inverse kinematics (IK). We then train the policy by setting the episodes' initial state to be the state of the grasping frame, including poses and joint velocity. During training, the initial state gradually moves backward in time until the desired starting state. For more details, please refer to the supplementary.

### 3.2 Continuous Dexterous Hand to Target Commercial Robots Evolution

A core hypothesis in our proposed paradigm of learning from human demonstration is the feasibility of interpolation between a human-hand-like five-finger dexterous robot and a target commercial robot. In this paper, we present solutions for dexterous-to-commercial-robot interpolation for two target commercial robots to show this. Since both the dexterous hand robot and the target robots are complicated in structure, how to design the evolution from the former to the latter is not straightforward and has a large design space. The design choice can significantly affect the policy transfer performance. Following [18], we first match the robot morphology between the dexterous and the target robot. With the same morphology, we then design a shared high-dimensional robot parameter space that allows defining an intermediate robot by simply setting these parameters. The evolution includes both the dexterous hand and arm.

**Dexterous Hand to Commercial Gripper Evolution** To match the number of fingers, we keep the thumb and gradually shrink the redundant fingers to be zero-size and zero-mass so that they eventually disappear. We also gradually change the range of the finger joints to match the desired

joint configuration. In addition, we add new position servo joints to the remaining fingers. The position servo joints are initially frozen and evolve to have the same range as the target fingers.

**Dexterous Arm to Commercial Robot Arm Evolution** A major challenge of arm evolution is the different ways of arm mounting. In simulation, the dexterous robot elbow joint is usually modeled as a virtual free joint that can move freely in the 3D space, while the commercial robot arms are usually mounted on fixed bases. Another challenge is the huge difference in the numbers of arm joints and joint physical parameters. Our solution is to attach the free joint of the dexterous arm to the end effector of the target robot and treat the entire dexterous robot as a *de facto* large gripper. During evolution, both the dexterous arm and the elbow free joint gradually shrink to zero so that the dexterous robot eventually evolves to be firmly attached to the target robot end effector.

Our robot evolution solution includes the changing of $D$ independent robot parameters such as body sizes and mass, and joint ranges and damping, where $D = 40$ for target Sawyer robot and $D = 42$ for target Jaco robot. The evolution process for Sawyer robot is illustrated in Figure 2.

### 3.3 Multi-dimensional Continuous Robot Evolution for Policy Transfer

As shown in Section 3.2, the evolution from a human-like dexterous robot to a target commercial robot is an extremely high-dimensional problem in robot parameter space (e.g. $D = 42$). This means there exist numerous choices of intermediate robot sequences that connect source to target robots. However, previous work REvolveR [18] assumes the same and uniform evolution progress hard-coded for all robot parameters and could be sub-optimal. We envision that when transferring the policy through intermediate robots, the optimal robot evolution should be flexible and able to smartly bypass difficult robot configurations to reach the target robot efficiently. In this section, we introduce an algorithm for automatically finding such optimized evolution strategy. We formulate the problem as a joint optimization of both robot evolution and the policy.

**Problem Statement** Suppose the source and target robot morphology is matched using the method in [18] or Section 3.2. Therefore the $D$ kinematics parameters of the source and target robots can be mapped to the same space denoted as $\boldsymbol{\theta}_S, \boldsymbol{\theta}_T \in \mathbb{R}^D$. Continuous function $F : [0, 1]^D \to \mathcal{M}$ defines an intermediate robot by interpolation between all pairs of kinematic parameters $\boldsymbol{\theta} = (\mathbf{1} - \boldsymbol{\alpha}) \odot \boldsymbol{\theta}_S + \boldsymbol{\alpha} \odot \boldsymbol{\theta}_T$ where $\boldsymbol{\alpha} \in [0, 1]^D$ is the *evolution parameter* that describes the evolution progress of each of the $D$ kinematics parameters. Given an expert policy $\pi^*_{F(\mathbf{0})}$ on the source robot $F(\mathbf{0})$, the overall goal is to find the optimal policy $\pi^*_{F(\mathbf{1})}$ on target robot $F(\mathbf{1})$. Note that the formulation of REvolveR [18] is a special case of our problem setting with $\boldsymbol{\alpha} = \alpha \cdot \mathbf{1}$ and $\alpha \in [0, 1]$.

**Differentiable Evolution Path Search (DEPS)** We adopt multi-dimensional robot evolution as a solution to the above problem as illustrated in Figure 3(a). Suppose there is a robot evolution path $\tau = (F(\boldsymbol{\alpha}_0), F(\boldsymbol{\alpha}_1), F(\boldsymbol{\alpha}_2), \ldots, F(\boldsymbol{\alpha}_{K-1}), F(\boldsymbol{\alpha}_K))$ where $\boldsymbol{\alpha}_0 = \mathbf{0}$, $\boldsymbol{\alpha}_K = \mathbf{1}$. In $k$-th phase, the policy optimization objective is to maximize the expected reward $\mathbb{E}[\rho^{\pi, F(\boldsymbol{\alpha}_k)}]$ on robot $F(\boldsymbol{\alpha}_k)$. For all $k$, the evolution parameter step size $\xi = ||\boldsymbol{\alpha}_k - \boldsymbol{\alpha}_{k+1}||_2$ is sufficiently small. In this way, we decompose the difficult robot-to-robot policy transfer problem into a sequence of $K$ easy problems.

The optimal solution to the above problem requires finding the optimal evolution path $\tau$ together with policy optimization. We propose an algorithm named Differentiable Evolution Path Search (DEPS) to find both the optimized robot evolution path $\tau$ in tandem with policy optimization. Given the current evolution parameter $\boldsymbol{\alpha}_k$, we aim to find the next best evolution parameter $\boldsymbol{\alpha}_{k+1} = \boldsymbol{\alpha}_k + \boldsymbol{l}_k$ as well as its optimal policy $\pi^*_{F(\boldsymbol{\alpha}_{k+1})}$. The optimization objective is formulated as

$$\max_{\boldsymbol{l}_k, ||\boldsymbol{l}_k||_2 = \xi} \quad \max_{\pi} \quad L = \mathbb{E}[\rho^{\pi, F(\boldsymbol{\alpha}_k + \boldsymbol{l}_k)}] - \frac{1}{2}\lambda||\mathbf{1} - (\boldsymbol{\alpha}_k + \boldsymbol{l}_k)||_2^2 \tag{1}$$

where $\lambda \in \mathbb{R}^+$ is a weighting factor. The first term optimizes the policy reward while the second term encourages the evolved robot to move close to the target robot. Since step size $\xi$ is small, we assume $L$ is locally differentiable w.r.t. $\boldsymbol{l}_k$. Then the direction of $\boldsymbol{l}_k$ can be estimated by

$$\boldsymbol{l}_k \approx \nabla_{\boldsymbol{\alpha}} L = \nabla_{\boldsymbol{\alpha}} \mathbb{E}[\rho^{\pi, F(\boldsymbol{\alpha})}]\Big|_{\boldsymbol{\alpha} = \boldsymbol{\alpha}_k} + \lambda(\mathbf{1} - \boldsymbol{\alpha}_k) = \boldsymbol{J} + \lambda(\mathbf{1} - \boldsymbol{\alpha}_k) \tag{2}$$

where $\boldsymbol{J} = \nabla_{\boldsymbol{\alpha}} \mathbb{E}[\rho^{\pi, F(\boldsymbol{\alpha})}]$ is the Jacobian of expected reward w.r.t. evolution parameter $\boldsymbol{\alpha}$. The Jacobian $\boldsymbol{J}$ can be estimated by finite difference with Monte-Carlo sampling of $\boldsymbol{\alpha}$. Concretely, we sample $n$ random vectors $\boldsymbol{\delta}_1, \boldsymbol{\delta}_2, \ldots, \boldsymbol{\delta}_n$ with $\ell^2$-norm of $\xi$. Suppose $\boldsymbol{\delta}_0 = \mathbf{0}$. For each $\boldsymbol{\delta}_i$, we

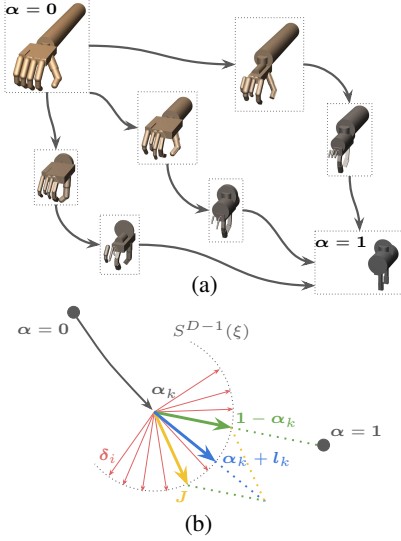

Figure 3: (a) Possible robot evolution paths from source to target robot. (b) Computation of the next evolution parameter $\alpha_{k+1} = \alpha_k + l_k$ in the path.

**Algorithm 1** Multi-dimensional Robot Evolution Policy Transfer

1: **Notation Summary:**
2: $\alpha \in [0,1]^D$: robot evolution parameter; $F : [0,1]^D \to \mathcal{M}$: continuous robot evolution function; $\pi_{F(0)}$: expert policy on the source robot; $\xi \in \mathbb{R}^+$: evolution step size; $q \in \mathbb{R}$: reward threshold; $\lambda \in \mathbb{R}^+$: weight factor; $\lambda_1 \in (0,1)$: shrink ratio;

---

3: $\alpha \leftarrow 0, \pi \leftarrow \pi_{F(0)}, l \sim S^{D-1}(\xi)$ // initialization
4: **while** $\alpha \neq 1$ **do**
5:     **if** $\mathbb{E}[\rho^{\pi, F(\alpha+l)}] < q$ **then**
6:         $\delta_1, \delta_2, \ldots, \delta_n \sim S^{D-1}(\xi), \delta_0 \leftarrow 0$ // sample $\delta_i$ vectors
7:         execute $\pi$ and $\rho_i \leftarrow \rho^{\pi, F(\alpha+\delta_i)}, \forall i \in \{0, \cdots, n\}$
8:         compute Jacobian $J$ from $\delta_j$ and $\rho_j$ according to Eq.(4)
9:         $l \leftarrow J/||J||_2 + \lambda(1-\alpha)/||1-\alpha||_2$
10:        $l \leftarrow l/||l||_2 \cdot \xi$
11:        **for** $e$ **in** $0, 1, \ldots, N_e$ **do**
12:           $\beta \sim \text{Uniform}(1 - \lambda_1^e, 1)$
13:           sample rollouts and train $\pi$ on robot $F(\alpha + \beta \cdot l)$
14:        **end for**
15:     **end if**
16:     $\alpha \leftarrow \min\{\max\{\alpha + l, 0\}, 1\}$ // make sure stay in $[0,1]$
17: **end while**
18: **return** $\pi$

execute $\pi$ on robot $F(\alpha_k + \delta_i)$ to get episode reward $\rho_i = \rho^{\pi, F(\alpha_k + \delta_i)}$. By the definition of Jacobian, if there were no noise, ideally we should have

$$J \cdot \delta_i = \rho^{\pi, F(\alpha_k + \delta_i)} - \rho^{\pi, F(\alpha_k)} = \rho_i - \rho_0, \quad \forall i \tag{3}$$

However, there is usually inevitable noise in the policy rollout and $\rho_i$. Let $\rho = [\rho_1, \rho_2, \ldots, \rho_n]^\top$ and $\Delta = [\delta_1, \delta_2, \ldots, \delta_n]$. The gradient can be estimated by Least Squares Minimization

$$J \approx (\Delta^\top \Delta)^{-1} \Delta^\top (\rho - \rho_0 \cdot 1) \tag{4}$$

Since the scale of the episode reward $\rho^{\pi, F(\alpha)}$ and therefore $J$ is task-dependent and can be arbitrary, to stabilize the evolution path search we scale both $J$ and $1 - \alpha_k$ in Equation (2) to be unit vectors to obtain the direction of evolution progression as follows

$$l_k = J/||J||_2 + \lambda(1 - \alpha_k)/||1 - \alpha_k||_2 \tag{5}$$

The next evolution parameter is obtained by $\alpha_{k+1} = \alpha_k + \xi \cdot l_k/||l_k||_2$, where $\xi$ is the evolution step size. The geometric interpretation of Equation (5) is illustrated in Figure 3(b).

We point out that the optimization objectives of our DEPS in Equations (2) and (5) can be viewed as a generalization of multiple previous works. When $\lambda \to +\infty$, the problem reduces to vanilla REvolveR [18] where $\alpha = \alpha \cdot 1$; when $\lambda \to 0$, the problem reduces to unrestricted robot and policy joint optimization such as [20] and [21]. Note that when $\lambda > 1$, $\{\alpha_k\}_k$ is guaranteed to converge to $1$, because it is easy to show for all $l$ with $\ell^2$-norm of $\xi$, $||1 - (\alpha_{k+1} + l)||_2 \leqslant ||1 - \alpha_k||_2 - \xi|\lambda - 1|$.

**Policy Optimization along the Evolution Path** Given the next evolution parameter $\alpha_{k+1} = \alpha_k + l_k$ and the optimized policy $\pi_{F(\alpha_k)}$, the next goal is to find the optimal policy $\pi^*_{F(\alpha_{k+1})}$ on robot $F(\alpha_{k+1})$. Instead of directly fine-tuning $\pi_{F(\alpha_k)}$ on $\alpha_{k+1}$, we adopt the following optimization objective with randomized evolution progression similar to [18] to achieve the goal:

$$\pi_{F(\alpha_{k+1})} = \arg\max_\pi \mathbb{E}_{\beta \sim \text{Uniform}(1-\lambda_1^e, 1)} \rho^{\pi, F(\alpha_k + \beta \cdot l_k)} \tag{6}$$

where $\lambda_1 \in (0,1)$ is the ratio of range shrinkage and $e \in \mathbb{N}$ is the iteration counter. The above optimization objective allows the policy to be fine-tuned on a set of randomly sampled robots within a range along the $l_k$ to improve the stability and robustness of training while also ensuring the sampled evolution parameter $\alpha$ to converge to $\alpha_{k+1} = \alpha_k + l_k$ when $e \to +\infty$. In practice, a batch of $\beta$ in Equation (6) is sampled in each iteration. The overall algorithm is illustrated in Algorithm 1.

## 4 Related Work

**Learning from Human Demonstration** Human demonstrations are often collected in terms of trajectories that could be obtained by teleoperation via virtual reality, motion capture markers, or kinesthetic control of robot. A policy is then learned by cloning the demonstrations [22, 23, 24, 25, 26, 27] or through inverse reinforcement learning [25, 28, 29, 30, 31]. In contrast to these methods, in our approach, the human does not control the robot to which the demonstration is transferred.

Human demonstrations need not always be in the form of state space information and a recent line of work rely on visual demonstration to enhance the scalability of data collection as it is much easier to watch humans than ask them to collect demos [32, 33, 34, 35, 36, 37, 38, 39, 40, 41]. Although in our case, the demonstrations are converted to the state space of the dexterous hand robot through inverse kinematics, they are originally collected by watching humans. Different from these prior works, we use the demonstration to transfer the policy to very different robots than the demonstrator.

**Learning Controllers for New Robot Morphology** Evolutionary methods to get new shapes date back to the early days of automata theory from Von Neumman [42]. In early 90s, Karl Sims showed the effectiveness of evolutionary methods in optimizing robot shapes and controllers [43, 44]. These works laid the foundation of the area where not only robots are optimized but also controllers are transferred across evolved robots. In recent years, many works have revived this idea using deep learning based graph evolution [45, 46, 47, 48, 49]. Our work does not treat the morphological evolution in the form of graph networks but rather gradually morphs the source robot to target robot.

**Relation to Path Planning** Our robot evolution path search problem is similar to the problem of path planning [50, 51, 52, 53, 54, 55]. However, unlike ordinary planning where the obstacles are known, in our problem the difficult robot configurations are unknown. Moreover, in ordinary path planning, evaluating the cost of a path is cheap, while in our case, transferring the policy to a new robot requires a large number of policy optimization iterations and is computationally costly.

## 5 Experiments

Our experiment is designed to verify the following two hypotheses: (1) our proposed DEPS algorithm can find optimized evolution paths in high-dimensional robot parameter space to improve policy transfer over vanilla linear evolution path [18]; (2) our HERD framework can learn from human demonstrations recorded in diverse modalities including sensor glove signal and visual data.

**Experiment Settings** We conduct experiments in MuJoCo physics simulation engine [56]. We adopt the five-finger dexterous hand robot provided in the ADROIT platform [57]. The high-fidelity models of the target robots and grippers are imported from robosuite environment [58] and have been used in previous robotics publications that eventually transferred their experiments to real robots such as [59, 60]. We use NPG [61] as the RL algorithm for policy optimization. Due to the nature of robot-to-robot policy transfer, the total number of RL iterations it takes to reach the goal of a certain success rate or episode reward cannot be set beforehand. So we instead report the number of policy training iterations and simulation epochs needed to reach 80% success rate on the tasks.

### 5.1 Learning from Sensor Glove Signal Demonstrations

**Tasks and Demonstrations** We use the three tasks from the Hand Manipulation Suite (HMS) [1]: `Hammer`, `Relocate`, and `Door`. In `Hammer`, the task is to pick up the hammer and smash the nail into the board; In `Relocate`, the task is to pick up the ball and move it to the target position; In `Door`, the task is to turn the door handle and fully open the door. We use a sparse reward setting where only task completion is rewarded. For each task, HMS [1] provided 25 human demonstrations recorded as palm and finger locations using a sensor glove named CyberGlove III. Then the expert policy on the dexterous robot is trained using DAPG [1].

**Qualitative Results** We provide visualization of the policies on the intermediate robots learned during evolution with our HERD in Figure 4. Starting from the dexterous robot, our proposed DEPS algorithm is able to find the next optimized evolved robot. The policy is also able to iteratively adapt to new evolved robots and successfully transfer to the target robots.

**Quantitative Comparison with Baselines** We compare our HERD against the following baseline methods for learning a policy on the target robot: (1) *From Scratch*: we train policy on the target robot from scratch with the RL algorithm under both sparse and dense shaped rewards; (2) *DAPG*

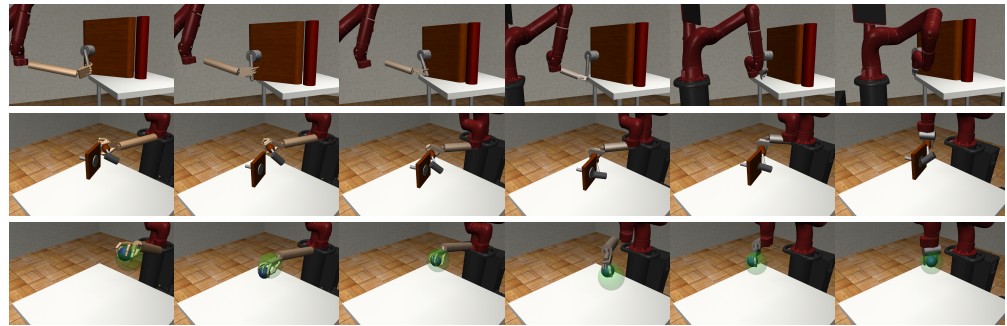

Figure 4: **Visualization of policies on evolving intermediate robots on Hand Manipulation Suite tasks [1].** We show the results of the learned policies on Door, Hammer and Relocate tasks in three rows respectively. Zoom in for better view.

| | Reward | Demo Agent | Door | | Hammer | | Relocate | |
|---|---|---|---|---|---|---|---|---|
| | | | # of sim epochs | # of train iters | # of sim epochs | # of train iters | # of sim epochs | # of train iters |
| From Scratch | Shaped | None | 608700 ± 79410 | 50725 ± 6617 | 248460 ± 9732 | 20705 ± 811 | 693540 ± 118860 | 57795 ± 9905 |
| | Sparse | None | ∞ | ∞ | ∞ | ∞ | ∞ | ∞ |
| DAPG [1] | Sparse | Sawyer | 245659 ± 24934 | 20471 ± 2077 | 161047 ± 14027 | 13420 ± 1168 | **114335 ± 15343** | 9527 ± 1278 |
| REvolveR [18] | Sparse | ADROIT | 273663 ± 28236 | 14302 ± 1843 | 110198 ± 4407 | 6356 ± 274 | 163611 ± 2859 | 8808 ± 450 |
| **HERD (Ours)** | Sparse | ADROIT | **201190 ± 23466** | **11229 ± 1166** | **91674 ± 6557** | **4434 ± 605** | 145158 ± 4207 | **8294 ± 1265** |

(a) Performance with Sawyer robot as the target robot.

| | Reward | Demo Agent | Door | | Hammer | | Relocate | |
|---|---|---|---|---|---|---|---|---|
| | | | # of sim epochs | # of train iters | # of sim epochs | # of train iters | # of sim epochs | # of train iters |
| REvolveR [18] | Sparse | ADROIT | 367612 ± 48859 | 17652 ± 2915 | 220169 ± 20262 | 11626 ± 509 | 259387 ± 2641 | 13215 ± 223 |
| **HERD (Ours)** | Sparse | ADROIT | **335196 ± 17938** | **13398 ± 2616** | **206214 ± 6576** | **8810 ± 390** | **214200 ± 5108** | **10179 ± 704** |

(b) Performance with Jaco robot as the target robot.

Table 1: **Experiments on Hand Manipulation Suite tasks [1].** The evaluation metrics is the number of simulation epochs and policy optimization iterations needed to reach 80% task success rate, shown as the "mean ± standard deviation" from runs with 5 random seeds.

[1]: a variant of NPG [61] with demonstration-augmented policy gradient. We use the rollouts of transferred HERD policy on the target robots as the demonstrations for DAPG; (3) *REvolveR* [18]: robot-to-robot policy transfer with continuous and linear robot evolution path which is a special case of our HERD. The results are illustrated in Table 1.

Training RL from scratch never completes the task to receive reward and is not able to learn on these tasks. Robot-to-robot evolution solutions with sparse rewards are able to successfully transfer the policy and even significantly outperform training from scratch with shaped rewards. Moreover, our HERD outperforms REvolveR [18], though the margin in simulation epochs is smaller than RL iterations in that HERD spends additional simulation epochs to obtain Jacobian as in Equation (4). We notice that our HERD also has advantage over DAPG trained using expert rollouts directly collected on the target robot. This shows that training a policy while maintaining sufficient reward/success during robot evolution has advantage over exploration from scratch even with demonstration.

## 5.2 Learning From Visual Demonstrations

**Task and Demonstrations** In addition to the demonstration captured by sensor glove, we are also interested in showing learning from visual human demonstration with our HERD. We use DexYCB dataset [2] as the source of human demonstration. DexYCB dataset is a multi-view RGB-D video dataset capturing human hand manipulating YCB objects [62]. We define the task as grasping and then moving the object of interest to the desired position within a distance threshold as demonstrated in the DexYCB videos. We use 5 YCB objects in Table 2. For each object, we take one video which results in 5 demonstrations in total. The human hand joint poses and object 6D poses are estimated by HRNet32 [63] and CozyPose [64] respectively. Then the expert dexterous policy is trained using the method in Section 3.1. The reward is sparse task completion reward.

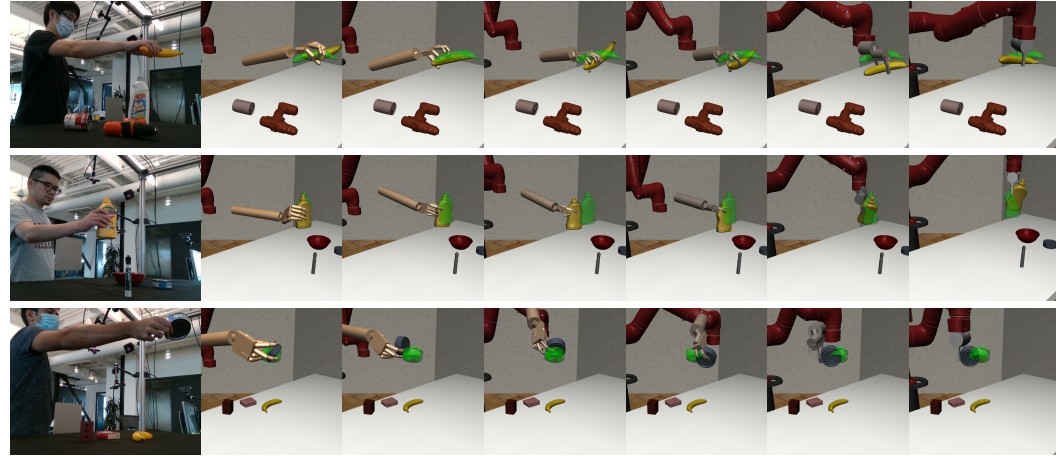

Figure 5: **Visualization of policies on evolving intermediate robots on DexYCB dataset manipulations [2].** The target robot is Sawyer. We show the results of manipulation policies learned from human demonstration on `011_banana`, `006_mustard_bottle` and `007_tuna_fish_can` in three rows respectively.

| YCB Objects | REvolveR [18] | | HERD (Ours) | |
|---|---|---|---|---|
| | # of sim epochs | # of train iters | # of sim epochs | # of train iters |
| `006_mustard_bottle` | $279563 \pm 20115$ | $25645 \pm 2181$ | $\mathbf{253920 \pm 22745}$ | $\mathbf{19361 \pm 1799}$ |
| `007_tuna_fish_can` | $320951 \pm 14466$ | $29812 \pm 1569$ | $\mathbf{286550 \pm 33816}$ | $\mathbf{23165 \pm 2693}$ |
| `008_pudding_box` | $306532 \pm 17388$ | $28537 \pm 1522$ | $\mathbf{264512 \pm 32775}$ | $\mathbf{22866 \pm 2986}$ |
| `011_banana` | $52094 \pm 2288$ | $4797 \pm 262$ | $\mathbf{51811 \pm 3490}$ | $\mathbf{4022 \pm 349}$ |
| `025_bowl` | $817255 \pm 46721$ | $73799 \pm 3892$ | $\mathbf{725919 \pm 59419}$ | $\mathbf{65957 \pm 5105}$ |

Table 2: **Experiments on the DexYCB dataset [2]**. The target robot is Sawyer. The evaluation metrics is the number of simulation epochs and policy optimization iterations needed to reach task success, shown as the "mean $\pm$ standard deviation" from runs with 5 random seeds.

**Result Analysis and Comparison** We compare against REvolveR [18] and illustrate the experiment results in Table 2. Our HERD outperforms REvolveR in terms of policy iterations as well as most reported simulation epoch metrics. One interesting finding is that the policy for manipulating `011_banana` object is the easiest task among all objects. A possible reason is that `011_banana` object has a geometry most suitable for both the dexterous hand and target gripper.

**Qualitative Results** We visualize the policies on the intermediate robots learned during evolution with our HERD in Figure 5. Again, our proposed DEPS algorithm is able to find the next optimized evolved robot while successfully transferring the policy to the target robot. For more details on the visualizations in this section, please refer to the supplementary video.

## 6 Conclusion

In this paper, we propose a framework named HERD for learning from human demonstration. HERD framework utilizes the continuous evolution of robots to transfer the expert policy from embodied dexterous hands to a target robot. We conduct experiments using the human demonstrations from Human Manipulation Suite [1] and DexYCB dataset [2]. We show that the proposed approach is able to transfer expert policy trained using human demonstrations in sensor glove signals or visual data to target two-finger Sawyer and three-finger Jaco robots, even with challenging sparse rewards.

**Limitations** We expect that our HERD would fail in learning from human demonstrations for robots with extremely different functions than humans, e.g. learning object manipulation policy for a robot arm mounted with a suction cup from human hand manipulation demonstrations. This is because our proposed human-to-robot continuous evolution solution can only handle different morphology but cannot handle different functions of the robot end-effector. We leave the exploration of more general human-to-robot continuous evolution solutions as future work.

**Acknowledgement**   This work is in part funded by JST AIP Acceleration, Grant Number JP-MJCR20U1, Japan. Deepak Pathak is supported by NSF grant IIS-2024594.

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
