# OpenReview forum: "HERD: Continuous Human-to-Robot Evolution for Learning from Human Demonstration"
_robot-learning.org/CoRL/2022/Conference — CoRL 2022 Poster_

### Official Review · Reviewer_Pofh · 2022-07-22

**Originality:** Fair
**Technical Quality:** Good
**Clarity Of Presentation:** Very Good
**Impact:** 3

**Recommendation:**

Weak Accept: I recommend accepting the paper, but will not argue for my recommendation if the majority of other reviewers have a different opinion.

**Summary:**

The authors present an approach for learning robot manipulation tasks from human demonstrations (HERD). Since the embodiments of the human demonstrator and that of the robot differ, the demonstration data cannot be utilized directly for robot learning. For overcoming this issue, a reinforcement learning agent learns to control a virtual robot which is step by step morphed from a human representation into the final robot representation. The agent's learning process is bootstrapped by using the demonstration data with the human embodiment. Subsequently, the embodiment is morphed gradually towards that of the robot. Every micro-update of the embodiment is followed by fine-tuning the agent's policy on the new embodiment. The main contribution of the paper is an algorithm for deciding how to shape the evolution path towards the final robot embodiment based on a Jacobian approximation.

**Issues:**

- Preliminaries: Here, you explain how REvolveR works, but at this point it is unclear how the method is connected to your approach. Therefore, the reader is wondering what the purpose of the explanation is. The connection to the method becomes clear in Section 3.3, but is confusing in Section 2. I recommend either adding a sentence (e.g., like "we extend this method" / "our approach is based on REvolveR") or moving the paragraph to another Section.

- Line 84: "These research progress" --> This research progress?

- Line 85: Figure 1(1)(2) looks strange. Just use Figure 1. The reader should be smart enough to read the caption and figure out on his own that only parts of the figure are relevant here.

- Regarding the approach, I was wondering if computing/approximating the gradient is actually necessary. Since you sample new embodiment parameters and do rollouts with each new parameter, why don't you simply pick the parameter set that yields the hightest reward instead of merging all parameter sets into a gradient based on the rewards? Referring to Figure 3, why not picking the best red arrow instead of computing a direction from all red arrows? If computing the Jacobian has advantages here it would be good if the authors could comment on it. It would also be possible to add this strategy as a baseline to the experimental validation.

**Quality Of The Limitations Section:**

Limitations are addressed clearly

**Reviewer Expertise:**

3: The reviewer is fairly confident that the evaluation is correct

**Robotics Focus:**

Highly relevant to robotics but no hardware experiments

**Strengths And Weaknesses:**

Strengths
- The authors show that their approach outperforms existing approaches due to the added freedom in adapting the embodiment parameters (even though the improved performance compared to REvolveR is rather small)
- The paper is written well and it is straightforward to read and understand

Weaknesses
- The approach is only applied to a single robot (Baxter). I think the paper would have been a lot stronger if the authors showed that multiple robots with different embodiments can successfully learn a task from the same human demonstration data.
- The tasks used for the experimental evaluation are very "kinematic". The authors do not discuss in the paper if the approach could also be applied to more "dynamic" tasks.


**Summary Of Recommendation:**

Overall, I think it is a solid paper without bigger weaknesses. However, I am not sure if the contribution of the paper justifies a publication at the top-tier CoRL conference. The improvements compared to previous state-of-the-art approaches (mainly REvolveR) seems to be rather small. Compared to REvolveR, the RL training seems to converge quicker to an acceptable task success rate (80% in the experimental evaluation). There is no empirical evidence that it also outperforms this algorithm. As mentioned in the other sections, the paper could be improved by applying the method to multiple different robots. Also, an ablation study would be helpful to justify design choices (for instance to evaluate the usefulness gradient computation, see my last point in the Issues-Section)

---

### Official Review · Reviewer_bvcA · 2022-07-30

**Originality:** Very Good
**Technical Quality:** Very Good
**Clarity Of Presentation:** Excellent
**Impact:** 3

**Recommendation:**

Weak Accept: I recommend accepting the paper, but will not argue for my recommendation if the majority of other reviewers have a different opinion.

**Summary:**

This paper proposes a method for transferring human skills in hand tool manipulation to a simple two-fingered robot. Specifically, a robot model with the same degree of freedom as a human first learns tool manipulation from a human demonstration, and then the robot model is gradually transformed into a two-fingered robot through an iterative process of reducing degrees of freedom and relearning policies. The main contributions of this study are the construction of a framework that integrates the process from human demonstration to tool manipulation by a two-fingered robot, and the reduction in the number of simulations required for training compared to existing methods.

**Issues:**

- Compared to the existing method REvolveR [18], it is not intuitive that the total number of simulations is reduced despite the increase in the dimensionality of the parameters involved in the evolution of the robot. In Table 1 and 2, the number of simulations used for policy learning and Jacobian computation should be broken down clearly and compared with REvolveR. The definition of simulation epoch should also be clarified.
- The task performance should be higher for hands with higher degrees of freedom. Therefore, in Eq.2, the Jacobian term (first term) is oriented toward the original dexterous robot hand. On the other hand, as the authors said, the lambda term (second term) is oriented toward the two-fingered hand. It is thought that there is a risk of them cancelling each other and stagnating in the middle of the learning process. Please add an explanation for this question.


**Quality Of The Limitations Section:**

Limitations are not well addressed

**Reviewer Expertise:**

5: The reviewer is absolutely certain that the evaluation is correct and very familiar with the relevant literature

**Robotics Focus:**

Highly relevant to robotics but no hardware experiments

**Strengths And Weaknesses:**

- The most significant contribution of this paper is that it provides a architecture from a human demonstration to task execution by a simple two-fingered robot. On the other hand, the weakest point of the paper is that the experiments were not conducted on a real robot.
- The tool manipulation task tested in the experiment has complex dynamics, especially contact and friction, which are difficult to simulate accurately and deviate significantly from the real world. The proposed method uses a simulator to learn policies, and it is unclear how well the learning results will perform in the real world.
- The evolutionary learning of the robot model is based on the existing method REvolveR [18]. The authors state that larger variation in evolutionary paths results in higher efficiency of learning, but they do not provide specific experimental data to support this.


**Summary Of Recommendation:**

The most significant contribution of this paper is that it shows a architecture from a human demonstration to task execution by a simple two-fingered robot, and it deserves to be accepted. However, due to the lack of mathematical novelty and the lack of real robot experiments, the evaluation was lowered one rank.

---

### Official Review · Reviewer_WPK2 · 2022-07-30

**Originality:** Fair
**Technical Quality:** Fair
**Clarity Of Presentation:** Good
**Impact:** 3

**Recommendation:**

Weak Accept: I recommend accepting the paper, but will not argue for my recommendation if the majority of other reviewers have a different opinion.

**Summary:**

This work introduces HERD, a framework for learning robotic policies from human demonstrations where the target robot has a different morphology (e.g. 2 fingered gripper) to the human (e.g. a hand). The framework has two main components. (1) Learning a robotic policy from human demonstrations where the robot policy has a similar morphology to the human demonstrations and (2) transfer the learned robotic policy to a target robot with a different morphology through continuous evolution.

The main contributions of this paper are as follows
* HERD, a framework for learning from human demonstrations by continuously evolving from source to target robots
* DEPS (Differentiable Evolution Path Search) which is a method for searching in the evolution parameter space to evolve from source to target robot
* Experiments in simulation which demonstrate that HERD can be used to transfer human demonstrations with sensor or visual observations to a target robot with a two-finger gripper.


**Issues:**

* Please add a discussion about how this work could be used to deploy policies on a physical robot.
* Please elaborate on the limitation raised
* Please address the points raised regarding referencing and comparing with REvolveR
* There are a number of grammatical errors. I give some examples below. Please review the paper for grammar.
   * Line 21: paradigm → paradigms
   * Line 83: demonstration → demonstrations
   * Line 84: these → this
   * Line 85: on → on a
   * Line 87: in → in a
   * Line 88: as → as in
   * Line 89: policy → policies
   * Line 90: demonstration → demonstrations
   * Line 240: “never completes the task to receive reward” this doesn’t make sense.
   * Line 246: training policy → training a policy


**Quality Of The Limitations Section:**

Additional details required

**Reviewer Expertise:**

3: The reviewer is fairly confident that the evaluation is correct

**Robotics Focus:**

Relevant but unlikely to deploy to hardware in near future

**Strengths And Weaknesses:**

Strengths
* This work address a relevant and important problem setting — how to learn from human demonstrations whilst also able to deploy the learned policies on robots with different morphologies.
* Simulation experiments are thorough, with relevant baselines
   * The authors test DEPS on 8 different tasks spanning two datasets and two state spaces (glove signals and visual demonstrations).
* The authors demonstrate that HERD can successfully be used to evolve policies from visual human demonstrations (the experiments on the DexYCB dataset). To my knowledge this is novel.
* The idea to use the Jacobian of the expected reward with respect to the evolution parameter to determine the evolutionary path is interesting.
* The paper is clearly written, and DEPS is well explained
* Related work appears good with one exception noted below
* The supplementary video is nice. I appreciate the time and effort the authors made to prepare a visual summary of the work. I especially appreciate the numerous examples of policies at different stages of the evolutionary process solving the different tasks.

Weaknesses
* I am not convinced that DEPS improves sample efficiency over REvolveR
   * In all cases the number of sim epochs to reach the same performance appears similar to REvolveR when compared with the reported standard deviation for DEPS.
      * I note that it is possible that the means have non overlapping confidence intervals and are significantly different.
   * I don’t think number of training iterations is a relevant metric for comparing policies since algorithms may differ substantilly in what occurs during a training iteration. If the authors intend to give a measure of training time I suggest using FLOPS or wall clock training time (although this is more relevant for a physical robot).
* The authors present a framework (HERD) but only demonstrate it on one source and target robot. I would be more convinced of the generality of their framework if they had demonstrated HERD on multiple target robots, especially given that evolution from the source to target robot in this paper appears to have been previously demonstrated in REvolveR.
* The learned policies are only deployed in simulation and there is no discussion of a path to deployment on a physical robot
* I appreciate the authors mentioning in the limitations section that HERD might fail if the target morphology was very different from the demonstrations. However, the limitations section is very brief, only 3 lines long. I would have liked to see more of a discussion as to why it might fail (e.g. is it impossible to define a continuous evolutionary path? Is it impossible to define MDPs with the same action space for source and target robots?). It would also have been really nice if this failure case could have been demonstrated experimentally.
* Limited algorithmic novelty and only moderate empirical novelty
   * HERD/DEPS is quite similar to REvolveR.
   * I note that the “core idea is to define a continuous interpolation between the dexterous hand robot and a target robot that have a different morphology” (line 41-42) was the main idea of prior work REvolveR.
   * Furthermore, successful evolution between the source and target robot has been demonstrated in prior work (REvolveR).
   * One way to improve the paper could be to provide a more in depth analysis of how DEPS differs from REvolveR. For example, does DEPS ever result in a noticeably different evolutionary path to REvolveR? Are there certain joints it favors vs REvolveR? Are there any circumstances under which REvolveR fails but DEPS succeeds? As of now it is not clear why I would choose to implement the more complex DEPS over REvolveR.
* Given that DEPS is similar to REvolveR and appears to build off it, I would have liked to see an explicit comparison with revolver in the related work.
   * The authors do cite REvolveR in the preliminaries, however REvolveR is absent from the related work
   * Additionally, in the last example in the supplementary video (tuna_fish_can object) where alpha is shared across all evolutionary parameters the authors refer to this as a special case of their framework. As I understand it, this is REvolveR and I think this should be acknowledged in the video.
   * Similarly HERD appears to be mostly a combination of existing techniques (DAPG / CozyPose + HRNet + Curriculum learning plus REvolveR) with a small modification to the Revolver algorithm. Combining pre-existing techniques can be a valuable contribution in itself, however I would have liked to see this more explicitly stated.


**Summary Of Recommendation:**

The paper addresses a relevant and interesting problem in robot learning, and the authors demonstrate the effectiveness of HERD on a number of different tasks in simulation. However at the moment the weaknesses outweigh the strengths of the paper.

---

### Official Review · Reviewer_mLRs · 2022-08-10

**Originality:** Fair
**Technical Quality:** Good
**Clarity Of Presentation:** Very Good
**Impact:** 3

**Recommendation:**

Weak Reject: I recommend rejecting the paper, but will not argue for my recommendation if the majority of other reviewers have a different opinion.

**Summary:**

The paper addresses the issue of robot learning from human demonstration with different structures. The proposed approach (named HERD) evolves a human hand into a robot hand, by simulating the evolutions in a dynamics simulator such as Gazebo. A policy is first learned from human demonstrations, and is transferred to the target robot through training on a sequence of intermediate simulated robots that evolve from the human hand to the target robot. At each evolution stage, the policy is updated and adapted to the new evolved hand by interacting in the simulator. To achieve this, the paper proposes to jointly optimise searching for optimal robot evolution paths and policy learned at each evolution.

The paper builds on top of a recently published work (REvolveR) using the same idea but with fixed hyper-parameters for guiding the search for evolution paths. Their extension to this approach performs significantly better on two benchmarks.

**Issues:**

1. How general is the matching process? For example, how easily can it be adapted to a 3-finger robot or even another 2-finger robot other than the Sawyer robot? In which case it should be easy to demonstrate on such robots since experiments are performed in simulations.
2. The approach would be stronger if instead of keeping the thumb and first finger and evolving the rest, we could randomly choose any 2 fingers to keep fixed and evolve the rest. At the very least an ablation study on this would shed light on the generalization of HERD.

**Quality Of The Limitations Section:**

Additional details required

**Reviewer Expertise:**

3: The reviewer is fairly confident that the evaluation is correct

**Robotics Focus:**

Highly relevant to robotics but no hardware experiments

**Strengths And Weaknesses:**

Strengths

HERD (and REvolveR) avoid the need statistical matching approaches that require the source and target to have similar kinematics and dynamics structures.

Weakness

While it doesn't require a human to match the state and action spaces of the source and target, it still requires human effort to "match the morphologies" of the two and "design a shared high-dimensional robot parameter space". The design of the robot parameter space still requires as much robotics expertise as the 'demonstration conversion' techniques the paper criticises.

**Summary Of Recommendation:**

Experimental results are not showing significant gains compared to REvolveR. In some cases REvolveR even performs better. This is in addition to the criticism I gave earlier, about the approach requiring expertise to match the morphologies and designing the shared robot parameter space. And this approach seems to be human-robot pair specific.

---

### Meta-Review · Area_Chair_jv56 · 2022-08-15

**Recommendation:** Accept (Poster)
**Confidence:** 4

**Metareview:**

Summary:

The paper addresses the issue of robot learning from human demonstration where the robotic component differs significantly from the human component in terms of morphology.  The mapping is done by simulation of evolution from one morphology to the other in simulation. The problem is cast as joint optimisation; concurrently optimising paths from one morphology to the other as well as learning a policy for the current morphology pairing.  Experiments are simulated.  The main contribution of the paper is an algorithm called DEPS, that learns the evolution path towards the final robot morphology.

The paper is incremental from the authors previous work, with middling originality and correspondingly limited impact.  The technical quality is good, and presentation is very good.  Limitations are not fully addressed.

Strengths
The problem tackled is relevant and important
The paper is based on an already high performing algorithm, and this contribution is similarly high-performing
The paper is well written and the algorithm well-explained
Accompanying videos are rich in information and support the paper well.

Weaknesses
An incremental publication containing a few main changes such as this may not be suitable for CoRL.
There is a lack of comparison to Revolver, which is also missing from the related work.
There is a lack of ablation studies to delineate the effects of the new additions to the algorithm.
No experiments on a real robot
Performance compared to the base algorithm seems similar an.
The benefits of DEPS seem minor, which needs more study and explanation.  Why is this a good approach?
The experimental section could be expanded to map to a range of different morphologies, which would make the paper more compelling, the results more powerful, and the analysis more interesting.


***Update***
The authors put a lot of work into improving the paper.  The paper still has several issues, however on balance the research would be of interest to the research community.